# Positive and Negative Affect Changes during Gender-Affirming Hormonal Treatment: Results from the European Network for the Investigation of Gender Incongruence (ENIGI)

**DOI:** 10.3390/jcm10020296

**Published:** 2021-01-14

**Authors:** Imke Matthys, Justine Defreyne, Els Elaut, Alessandra Daphne Fisher, Baudewijntje P. C. Kreukels, Annemieke Staphorsius, Martin Den Heijer, Guy T’Sjoen

**Affiliations:** 1Department of Endocrinology, Ghent University Hospital, 9000 Ghent, Belgium; Guy.TSjoen@ugent.be; 2Center for Sexology and Gender, Ghent University Hospital, 9000 Ghent, Belgium; Els.Elaut@ugent.be; 3Andrology, Women’s Endocrinology and Gender Incongruence Unit, Department of Experimental, Clinical and Biomedical Sciences, University of Florence, 50139 Florence, Italy; alessandra.fisher@gmail.com; 4Department of Medical Psychology, Amsterdam University Medical Center, location VUmc, 1081 HV Amsterdam, The Netherlands; b.kreukels@amsterdamumc.nl; 5Center of Expertise on Gender Dysphoria, Amsterdam University Medical Center, location VUmc, 1081 HV Amsterdam, The Netherlands; a.staphorsius@amsterdamumc.nl (A.S.); m.denheijer@amsterdamumc.nl (M.D.H.); 6Department of Endocrinology, Amsterdam University Medical Center, location VUmc, 1081 HV Amsterdam, The Netherlands

**Keywords:** affect, quality of life, gender-affirming hormonal treatment, PANAS questionnaire, estrogens, testosterone, antiandrogen therapy, transgender

## Abstract

Improving transgender people’s quality of life (QoL) is the most important goal of gender-affirming care. Prospective changes in affect can influence QoL. We aim to assess the impact of initiating gender-affirming hormonal treatment (HT) on affect. In the European Network for the Investigation of Gender Incongruence (ENIGI) study, we prospectively collected data of 873 participants (451 transwomen (TW) and 422 transmen (TM)). At baseline, psychological questionnaires including the Positive and Negative Affect Schedule (PANAS) were administered. The PANAS, levels of sex steroids and physical changes were registered at each follow-up visit during a 3-year follow-up period, starting at the initiation of hormonal therapy. Data were analyzed cross-sectionally and prospectively. Over the first three months, we observed a decline in positive affect (PA) in both TM and TW. Thereafter, PA reached a steady state in TW, whereas in TM there was also a second decline at 18 months. In both TM and TW there was no persisting difference comparing baseline to the 36-months results. Concerning negative affect (NA), we observed a decline during the first year in TM, which sustained during the second year and was not different anymore at 36 months compared to baseline. In TW though, we did not find any change of NA during the entire follow-up. Even if some of these results show significant differences, they should be considered with caution, since there was no control group and the absolute differences are small. No association between affect and the level of sex steroids was observed. Baseline QoL and psychological burden are related to affect independently from gender but are not necessarily good predictors of the evolution of one’s affect during the gender-affirming process. Further research is necessary to investigate these preliminary results.

## 1. Introduction

Transgender is a term that describes people whose gender identity differs from the sex they were assigned at birth. Transgender people can experience great distress due to this incongruence, also known as gender dysphoria [1]. Gender-affirming care may include social transition, hormonal therapy and gender-affirming surgery. Hormonal therapy in transgender men (TM) consists of testosterone agents, usually administered intramuscular or transdermal. The current hormonal therapy for transgender women (TW) involves estrogens (administered orally or transdermally) and antiandrogens (to suppress testosterone levels and decrease masculine secondary sexual characteristics). To maintain virilization in TM and feminization in TW, hormonal therapy is continued life-long. Except in TW, if orchiectomy is desired, antiandrogens can be discontinued postoperatively [2,3].

Research has shown that gender-affirming therapy generally leads to increased quality of life (QoL) [4,5]. However, it is currently unknown what the individual contribution of the different treatment modalities is to the increase of QoL. As gender-affirming care is highly individualized, a person may undergo one or more gender-affirming medical interventions. Hormonal therapy is often considered as one of the first steps, as it is the most widely accessible option. Understanding the impact of hormonal therapy on QoL is hence crucial.

QoL is an umbrella term for a range of different physical and psychosocial domains [6]. Several factors such as perceived discrimination, amount of social support, and level of medical transition influence QoL in transgender people. From previous studies we know that transgender people suffer more from psychiatric complaints (especially problems with mood, anxiety and substance use/abuse) compared to cisgender individuals [7,8]. Depression and anxiety complaints are characterized by high negative affect (NA), which corresponds to a high level of distress and unpleasurable feelings [9,10]. Affect refers to the outward expression of feelings and emotions. Positive affect (PA) is generally described as the experience of pleasurable emotions [11]. The interest in PA has increased since evidence arose that PA is correlated with better health prospects [12].

At this point though, little is known about the affectivity of transgender people and its evolution during the gender-affirming process. Previous data by Fisher et al. showed that hormonal therapy decreases depressive symptoms and improves general psychopathology in transgender people [13]. Hence, we hypothesized that affect is influenced by hormonal therapy, leading to an increase in PA and a decrease in NA. We therefore conducted a multicenter trial in a transgender population where we assessed negative and positive affect separately from the initiation of hormonal treatment onwards. More specifically, we wanted to describe the effect of testosterone or estrogens with antiandrogens on affect. In addition, we aimed to maintain a more holistic view on the evolution of affect during the gender-affirming process by identifying other possible factors such as sexual desire, physical changes, QoL, psychological burden and gender dysphoria.

## 2. Experimental Section

### 2.1. Study Population

Before visiting the endocrinology department, transgender people first visited mental health specialists from the participating clinics. More information on the psychological part of the ENIGI (European Network for the Investigation of Gender Incongruence) initiative was previously published [14]. Transgender persons were subsequently referred to the endocrinology department. In total, 1669 people were included in the endocrine part of the ENIGI study, which started in 2010 (1055 in Amsterdam, 357 in Ghent, 67 in Florence and 190 in Oslo). The study protocol of the endocrine part of ENIGI was also previously published [3]. Written informed consent was obtained according to the institution’s guidelines. Positive and negative affect were prospectively measured with the Positive Affect Negative Affect Schedule (PANAS) [15], which was added to the battery in September 2012. In Ghent, 93 participants did not complete the questionnaire, as they were included in the ENIGI protocol before the introduction of this questionnaire. Participants in Oslo did not participate in this study arm. Due to data lock, the number of people included in this analysis decreased over time (Figure 1 and Figure 2). Data from 708 participants were entered into the database in Amsterdam (of which 117 participants did not complete the questionnaire at baseline of the endocrine part), whereas in Ghent and Florence, data were entered from all participants who completed the survey (Ghent: *n* = 248, 16 participants did not fill in the questionnaire at baseline of the endocrine part; Florence: *n* = 34, 33 participants did not fill in the questionnaire at baseline of the endocrine part). In total, 873 participants who completed the questionnaire at baseline of the endocrine part were included in this prospective analysis (451 TW and 422 TM). Characteristics of the study population at baseline are shown in Table 1.

### 2.2. Gender-Affirming Hormone Therapy

After filling out baseline questionnaires, gender-affirming hormone therapy was initiated according to the ENIGI study protocol, in accordance with the World Professional Association for Transgender Health (WPATH) Standards of Care, edition 7 [16]. In Ghent, TM received intramuscular long-acting testosterone undecanoate (Nebido^®^ 1000 mg, Bayer, Diegem, Belgium, once every 12 weeks). In Amsterdam and Florence, treatment options for TM included testosterone gel in a daily dose of 50 mg and intramuscular administration, either as testosterone esters (Sustanon^®^ 250 mg Aspen, Dublin, Ireland, every 2 weeks) or testosterone undecanoate (Nebido^®^ 1000 mg Bayer, Diegem, Belgium, every 12 weeks). In TW, estrogens plus antiandrogens are administered. Antiandrogen therapy consisted of cyproterone acetate 25 to 50 mg once daily (Androcur^®^ Bayer, Diegem, Belgium and Androcur^®^ Bayer, Mijdrecht, the Netherlands). Estrogen therapy generally consisted of estradiol valerate 2 mg (Progynova^®^ Bayer, Diegem, Belgium and Progynova^®^ Bayer, Mijdrecht, the Netherlands) twice daily. In patients older than 45 years of age, estradiol was administered transdermally in the form of estradiol patches (Dermestril^®^ Besins, Belgium, or Systen^®^ Bayer, Mijdrecht, the Netherlands) in a dose of 100µg/72 h to avoid the increased risk for thrombosis from oral estrogens caused by the first pass effect of the liver. In case of intolerance, estrogens were administered as gel (Estrogel^®^, Besins, Belgium) in a dose of 1.5 mg twice daily.

### 2.3. Other Medication

Participants were asked which other medication they were taking at each study visit. This data was confirmed by the treating physician. For the current study, people using mental health medication including antidepressants and benzodiazepines were identified at each study visit for later analysis.

### 2.4. Questionnaires

#### 2.4.1. PANAS

The Positive and Negative Affect Schedule (PANAS) (in Dutch, French and Italian) is a validated survey that was constructed to assess positive and negative valanced emotional states and attitudes. Positive Affect (PA) comprises feelings of enthusiasm, concentration, activity, alertness and pleasurable engagement, whereas low PA is characterized by sadness and lethargy. Negative Affect (NA) reflects the extent to which a person experiences subjective distress and unpleasurable engagement, resulting in feelings of anger, contempt, disgust, guilt, fear, nervousness. Although the terms PA and NA might suggest that these two mood factors are opposites, they are highly distinctive dimensions. The questionnaire consists of twenty questions: ten assessing PA, ten assessing NA. All questions are answered on a 4-point Likert scale, ranging from “very slightly or not at all” (1) to “extremely” (4). The PANAS was assessed at baseline and during each endocrinological follow-up moment. Internal consistency in the current sample was high for both PA (Cronbach’s alpha = 0.951) and NA (Cronbach’s alpha = 0.912) [15,17].

#### 2.4.2. Sexual Desire Inventory (SDI)

The Sexual Desire Inventory (SDI) is one of the most frequently used instruments to evaluate sexual desire [18]. The SDI consists of 14 items measured across two dimensions: dyadic sexual desire and solitary sexual desire. Total sexual desire is calculated by adding the scores of both dimensions. A higher total score reflects a higher level of desire. The term ‘dyadic sexual desire’ refers to the desire for sexual intimacy with another person, whereas ‘solitary sexual desire’ refers to the desire to engage in sexual behavior by or with oneself. This may involve a wish to refrain from intimacy with others, although both dimensions can also be high in the same person. Previous use of the SDI in the current sample is discussed in Defreyne et al. [19].

#### 2.4.3. Psychological Battery Assessed Only at Baseline

The Utrecht Gender Dysphoria Scale (UGDS) (in Dutch, French and Italian) is a scale that consists of 12 questions answered on a five-point Likert scale, ranging from ‘completely agree’ to ‘completely disagree’. This questionnaire was used to measure the degree of experienced gender dysphoria. A higher score indicates stronger gender dysphoria. Internal consistency of the UGDS in the current sample was good (Cronbach’s alpha = 0.797) [20].

The Symptom Checklist 90-Revised (SCL-90R) (in Dutch, French and Italian) is a questionnaire assessing self-reported psychological burden on eight symptom scales: somatization, sleeping problems, paranoid ideation/psychoticism, agoraphobia, depression, hostility, anxiety, interpersonal sensitivity and a global score ‘overall psychoneurotic distress’, as previously described [21]. A higher score indicates more psychological burden. Internal consistency in the current sample ranged from good to high for all factors (somatization: α = 0.797, sleeping problems: α = 0.761, paranoid ideation/psychoticism: α = 0.868, agoraphobia: α = 0.822, depression: α = 0.924, hostility: α = 0.805, anxiety: α = 0.882, interpersonal sensitivity: α = 0.925, overall psychoneurotic distress: α = 0.760).

Life as a whole (Bradburn) (in Dutch, French and Italian) is a questionnaire assessing general and social quality of life. We refer to this scale when we report QoL. General quality of life consisted of four questions, social quality of life consisted of eight questions, answered on a three-point Likert scale with “yes”, “more or less” and “no” as possible answers. A higher score indicates worse quality of life [22].

### 2.5. Physical Changes

#### 2.5.1. Ferriman and Gallwey Classification

The modified Ferriman and Gallwey Classification was used to subjectively assess the degree of male hair growth, only terminal hair growth is considered in the scoring. Nine sites (lip, chin, chest, upper back, sacroiliac region, upper abdomen, lower abdomen, arm and medial thigh) are graded according to the following: 0 = none, 1 = slight, 2 = moderate, 3 = dense and 4 = very dense. The Ferriman and Gallwey score has a minimum value of 0 and a maximum of 36 [23].

#### 2.5.2. Side-Effects Questionnaire

This investigator-designed (nonvalidated) questionnaire to assess side-effects of HT was used to evaluate the persistence of vaginal bleeding and spotting in TM. Vaginal bleeding and spotting intensity were scored on a scale ranging from 0 to 3, with 0 indicating “no”, 1 indicating “mild”, 2 indicating “moderate” and 3 indicating “severe” spotting or bleeding. This was self-reported by participants, without quantification by a physician.

#### 2.5.3. Gender-Affirming Surgery

Data on chest surgery, gonadectomy or genital gender-affirming surgery were logged at each visit. For those who underwent surgery during the investigated time period, the median time to chest surgery was 17.0 months (P25–P75: [12.0–17.0]) months in TW and 15.5 [5.8–28.8] months in TM. Median time to gonadectomy was 14.0 [11.0–18.3] months in TW and 15.0 [12.0–19.0] months in TM. Median time to gender-affirming genital surgery was 19.0 [16.3–24.0] months in TW and 21.5 [11.7–24.8] months in TM.

### 2.6. Laboratory Analyses

In Ghent, Amsterdam and Florence, laboratory analyses were performed using commercially available immune-assays, as previously described in Wiepjes et al. [24] and Defreyne et al. [2,19]. However, in Amsterdam, estradiol was measured using a LC-MS/MS after July 2014 (AUMC, location VUmc, Amsterdam, the Netherlands) with an interassay CV of 7% and a LOQ of 20 pmol/L.

### 2.7. Statistical Analyses

Data were analyzed prospectively and cross-sectionally. We attempted to construct a model for prospective changes in positive and negative affect using mixed models analyses, with serum levels of sex steroids as a covariate and undergoing surgery as a factor, but this was not possible, as the data were skewed and nontransformable. Cross-sectional data were analyzed using analyses of covariance (ANCOVA). Prospective data were analyzed using Friedman’s test or Wilcoxon’s signed rank test for continuous non-normally distributed data. For categorical variables, the difference between prospective scores between categories was assessed by Mann–Whitney U test or Kruskal–Wallis H test.

To control for differences in testosterone or estrogen mode of administration and laboratory analyses of serum sex steroid levels, all statistics were retested in groups using the same type of treatment as well as groups in whom serum sex steroid levels were analyzed using the same method.

For normally distributed data, values are shown as mean ± standard deviation (SD), for non-normally distributed data, values are shown as median [percentile 25–percentile 75]. The significance level was set at *p* < 0.05. All tests were two sided. If required, a Bonferroni–Holm correction was applied to adjust for multiple comparisons [25], which explains why some *p*-values <0.05 are not being marked as significant.

## 3. Results

### 3.1. Affect through the Course of Hormone Therapy

#### 3.1.1. Positive Affect

Median total PA scores decreased over the first 3 months in TW, from median [percentile 25–percentile 75] 33.0 [28.0–38.0] to 31.0 [25.0–37.0] (mean Δ −1.98 (95%CI −3.13 to −0.82), *p* = 0.001). Thereafter, PA remained stable until the end of the follow-up compared to the previous follow-up visit (all *p* = NS), but stayed significantly lower compared to baseline at all follow-up visits, except at 36 months, when there was no significant difference anymore compared to baseline.

Median total PA scores also decreased over the first three months in TM, from 33.0 [28.0–38.0] to 28.0 [23.0–33.0] (mean Δ −4.49 (95%CI −5.58 to −3.40), *p* < 0.001), with a second decrease between 3 and 18 months, to 25.0 [18.5–31.5] (mean Δ −2.78 (95%CI −5.04 to −0.51, *p* = 0.017). Thereafter, PA remained stable until the end of the follow-up compared to the previous follow-up visit (all *p* = NS), but at 36 months there was no significant difference anymore compared to baseline.

#### 3.1.2. Negative Affect

At baseline, the median NA scores for TW were 15.0 [12.0–19.0]. Median total NA scores remained constant in TW (all *p* = NS). Median total NA scores decreased over the first year in TM, from 14.0 [12.0–19.0] to 12.0 [11.0–16.0] (mean Δ −2.23 (95%CI −3.08 to −1.38), *p* < 0.001), remaining stable thereafter (all *p* = NS), but there was no significant difference anymore at 36 months compared to baseline.

#### 3.1.3. Subgroup Analysis

A comparison was performed between people with prospective increasing PA or NA scores and people with stable or decreasing scores. The three PA as well as NA groups (increasing, stable or decreasing) did not differ regarding Ferriman and Gallwey scores, severity of spotting or menstruation and having surgery (chest surgery, gonadectomy or gender-affirming genital surgery) performed compared to baseline, at any of the follow-up visits (all *p* = NS).

### 3.2. Differential Effect of Feminization versus Masculinization

Prospective changes in PA and NA did not differ between TW and TM at any of the study visits (all *p* = NS).

Prospective changes in PA and NA were not correlated with prospective changes in serum levels of sex steroids (all *p* = NS), independent of type of HT.

### 3.3. Physical Correlates of Affect

Prospective changes in PA were correlated to prospective changes in total Ferriman and Gallwey scores in TM (ρ = 0.910, *p* = 0.002).

Prospective changes in total PA and NA scores did not differ in groups with versus without spotting or vaginal bleeding, nor were there any associations with prospective changes in the severity of spotting or vaginal bleeding.

TW who underwent vaginoplasty showed a prospective increase in NA over 2 years (mean +2.00, 95% CI −3.17 to 7.17), whereas those without vaginoplasty showed a prospective decrease (−4.86, 95% CI −10.69 to 0.98), *p* = 0.020).

The use of mental health medication including antidepressants and benzodiazepines was not associated with prospective changes in total PA or NA scores in both TW and TM (all *p* = NS).

### 3.4. Psychological Correlates of Affect

#### 3.4.1. Cross-Sectional Analyses

At baseline, higher total PA in TW was associated with lower gender dysphoria scores, higher total and social QoL and lower self-reported psychological burden (SCL-90R scores). Higher baseline NA scores in TW were associated with worse baseline total and social QoL, a higher self-reported psychological burden and higher sexual desire scores. Over follow-up, PA was positively associated with baseline QoL at 3 and 6 months and negatively associated with baseline psychological burden at 3 and 12 months. NA was positively associated with higher baseline psychological burden at 3, 6 and 12 months and positively associated with higher sexual desire scores at 3 months (Table A1).

In TM, higher baseline PA was associated with higher sexual desire scores. Higher baseline NA was associated with worse baseline total and social QoL and higher baseline self-reported psychological burden. Over follow-up, PA was positively associated with higher sexual desire at 3 months. Higher NA scores were associated with worse baseline total QoL at 3 and 6 months and a higher baseline self-reported psychological burden at 3, 6 and 12 months (Table A1).

#### 3.4.2. Prospective Analyses

Prospective changes in PA scores were positively associated with higher baseline total QoL scores in TW after 6 months of HT. No other significant associations were observed.

## 4. Discussion

Affect is an important indicator of psychopathology and influences QoL [11]. To our knowledge, there are currently no data on the influence of HT on changes in affect over the course of the gender-affirming process. Our study emphasizes a holistic view of the evolution of affect during HT. Therefore, we conducted the first prospective study to explore this effect in a real-life cohort of 873 transgender people. We did not select or exclude specific groups (e.g., those with co-occurring psychological/psychiatric difficulties).

In our study, we showed that PANAS PA scores decreased the first three months after the initiation of gender-affirming hormonal therapy in both TM and TW. This evolution is the opposite of the change of gender dysphoria after initiation of hormonal therapy described in the study of Fisher et al. [13]. However, PA reflects emotions, often from social situations, and can thereby be strongly influenced by the social difficulties transgender people experience. In the study of Fisher et al. social indicators also worsened across time [13]. Other than that, the initial decrease of PANAS PA scores from baseline could be explained by a reduction of the initial excitement about the start of the often long-awaited therapy or simply by the emotional upheaval during the first months of HT. In TM we also saw a smaller second decline in PANAS PA scores between three and eighteen months of therapy, possibly due to the fact that physical changes after initiating HT only occur after a certain amount of time [26]. This is supported by the positive correlation we found between PA scores and Ferriman and Gallwey scores in TM.

To assess the role of sex steroids on these transient changes, we compared the PANAS scores between TM and TW during HT. We did not find a contrasting evolution. Moreover, it seems that both sex steroids influenced affect in a similar way. We did not find any correlation between the changes of affect and the level of sex steroids. Similar findings were observed in cisgender women, where no association was found between self-reported NA and sex hormone levels across the menstrual cycle [27]. This is in line with our finding that the PANAS NA scores remained stable over time in TW. Interestingly, we did notice different perceptions in our clinical practice. TM experience less abundant emotions after initiation of hormonal therapy but TW, on the other hand, claim to experience more intense emotions in particular situations. Hence, we believe that these different perceptions are not solely due to HT and that feminizing HT does not contribute to extended emotionality.

When we took a closer look at the role of testosterone, we hypothesized that testosterone may have protective effects for affective problems characterized by high NA such as depressive complaints and anxiety. This hypothesis was strengthened by the observation that PANAS NA scores in TM decrease significantly during the first year of treatment. However, we did not find an association between the serum level of testosterone and affect nor an association between the masculinizing effect of hormonal therapy and affect, regardless of the type of hormonal therapy. This finding contradicts previous assumptions in the literature that the route of administration of testosterone is important. For instance, transdermally administered testosterone was suggested to be more effective in improving mood complaints than intramuscular injections [28].

Body modifications effectuated by HT are an important therapy goal for transgender people, as they may experience a high level of body uneasiness at start of the transition. That is why we wanted to explore the effect of physical changes on affect during transition. From our findings, the role of physical changes on affect is not straightforward. We observed that TM who still experience spotting after initiation of hormonal therapy do not report a higher PANAS NA score. Also, self-reported PA in TM increased with an increasing Ferriman and Gallwey score. Unfortunately, we could not retain a difference in the Ferriman and Gallwey score in the subgroup analysis of PA (decreasing, increasing or stable) due to small sample size in one of the groups. Secondly, we observed that TW who had undergone a vaginoplasty reported an increase in PANAS NA scores during the first 2 years after surgery. At first sight this may seem an unexpected finding, but this short-term increase may be explained by the distress related to recovery and possible complications after surgery, but more long-term data are needed. On the other hand, TW who chose a vaginoplasty consistently showed lower total PANAS NA scores than TW who did not. This suggests that there are other factors influencing NA of these TW. For example, we need more insight in the motives for gender-affirming genital surgery to interpret these findings correctly.

Regarding the psychological correlates, the cross-sectional and prospective results were variable. The PANAS PA scores were only positively associated with baseline QoL and inversely associated with baseline gender dysphoria and self-reported psychological burden in TW. While the PANAS NA scores were positively associated with baseline self-reported psychological burden and inversely with baseline QoL, independently from gender. In addition, the only prospective finding was a positive association between PANAS PA scores and baseline QoL in TW after six months of hormonal therapy. No other prospective associations were observed, even when corrected for the use of mental health medication. Further research is necessary to investigate the role of affect and the prospective changes in QoL, psychological burden and gender dysphoria as we assessed the psychological battery only at baseline. For now, we can conclude that baseline QoL and psychological burden are related to affect but are not necessarily predictive for the evolution of someone’s affect.

Remarkable is the difference regarding the association between affect and sexual desire when comparing TM and TW. In TW, the higher PANAS NA scores, the higher the total sexual desire score. On the contrary, we observed that in TM it is not the PANAS NA score, but the PA score that correlates with higher sexual desire. This finding could be explained by a common observation in our clinical practice, namely that TW may sometimes experience sexual desire as distressing, for example, due to unexpected erections. In TM, on the other hand, sexual desire is often found less distressing because they rapidly progress to the male expectation pattern, which is high sexual desire, after initiation of testosterone. The current data seem to be in line with the findings of previous retrospective studies and a previous study of the ENIGI group [19,29]. The previous results from the ENIGI group showed an initial decrease in sexual desire in TW and an initial increase in TM. However, 3-year follow-up data revealed sexual desire levels to be higher than baseline levels in TW and comparable to baseline levels in TM [19]. More information on dyadic and solitary sexual desire and the PANAS can be found in Defreyne et al. [19].

The results of our study should be considered in the light of some limitations. Firstly, we did not include a control group. Secondly, objective measurement for feminization, like 3D evaluation of breasts, was not available. Thirdly, the PANAS questionnaire is exclusively self-reported, which makes it difficult for a researcher to interpret one person’s score as compared to the score of someone else’s. Also, it only evaluates the emotions of a particular moment, which may be very dependent on unknown external life events or worries. Another contributing factor to possible bias was that we were not able to correct for transgender people who are not able to illustrate their emotions, since this autistic trait is found to be more prevalent among the transgender population than in the general population [30]. Lastly, we observed that some patients did not always fill out the repeat PANAS questionnaires veraciously. In the literature, we did not find another research group who had encountered the same repetition bias of the PANAS questionnaire. Unfortunately, we did not include questionnaires on minority stress in the prospective part of the ENIGI endocrinology study. We hypothesize that minority stress during the gender-affirming process may explain the decrease in PA scores after the initiation of HT. The physical effects of initiating HT may not live up to expectations during the first months of HT, which may lead to people experiencing even more minority stress initially, especially when mental health coaching is not easily accessible.

Strengths of this study are the prospective design, large sample size and its novelty. Our study cohorts are well-defined and participants adhered to a strict treatment regimen.

## 5. Conclusions

As the first prospective study on affect and gender-affirming hormonal therapy, this study provides novel insights. We found that PANAS PA and NA scores temporarily decrease after initiation of gender-affirming hormonal treatment, except for the PANAS NA score in TW that remains stable during the entire follow-up.

A remarkable difference was observed with regards to the relation between sexual desire and affect in TM and TW. While in TM it is the PANAS PA score that positively correlates with sexual desire, it is the PANAS NA score in TW. Despite these findings, we did not find any association between sex steroid levels and affect nor a convincing association with physical changes. Baseline QoL and psychological burden do not tend to be the perfect predictors for the evolution of affect during the transitional process, as the results were variable.

We can conclude that measuring the impact on affect during a gender-affirming process is complex and that sex steroids alone do not account for significant changes in affect because of its intertwined nature. Although we found several statistically significant differences in the PANAS scores across time, these differences should be considered with caution since they are small in absolute values.

Overall, the results of this study should be seen as a guidance to generate hypotheses for further research. In this regard, the PANAS does not seem to be a useful tool in practice to evaluate gender-affirming treatment in a transgender population without taking the whole societal context (e.g., minority stress, feelings of gender dysphoria, changing interpersonal relations after initiating HT, etc.) into account.

## Figures and Tables

**Figure 1 jcm-10-00296-f001:**
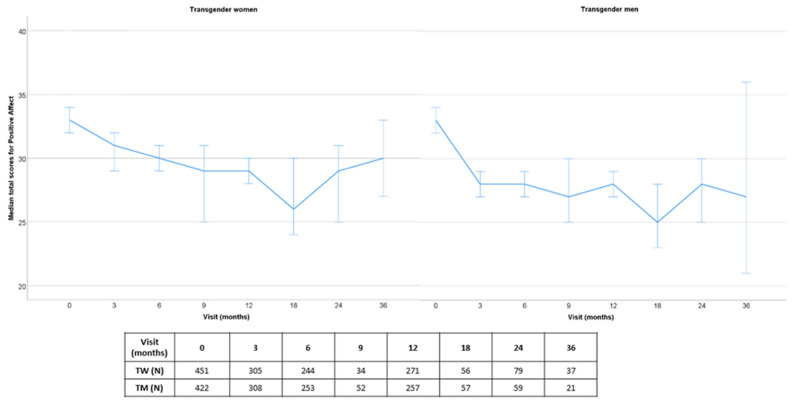
Prospective median total scores for positive affect over 36 months of follow-up (with error bars—95% confidence interval), subdivided by gender identity.

**Figure 2 jcm-10-00296-f002:**
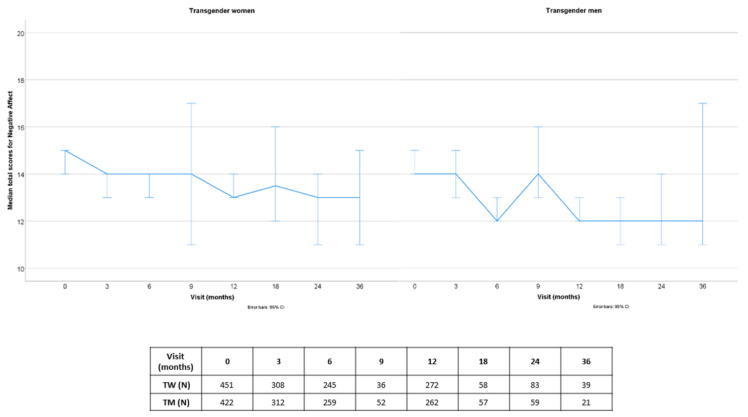
Prospective median total scores for negative affect over 36 months of follow-up (with error bars—95% confidence interval), subdivided by gender identity.

**Table 1 jcm-10-00296-t001:** Baseline characteristics of the study population subdivided by gender identity.

			TW (*n* = 451)	TM (*n* = 422)
Age (years)			27.0 [22.0–41.0]	22.0 [20.0–28.0]
Baseline PA scores			32.5 [27.0–37.0]	33.0 [27.0–38.0]
Baseline NA scores			14.0 [12.0–19.0]	14.0 [12.0–19.0]
Gender-affirming hormonal therapy	Testosterone (60 missing)	AG 25 mg once daily		18 (5.0%)
		AG 50 mg once daily		106 (29.3%)
		TU 1 g once every 12 weeks		121 (33.4%)
		TE once every 2 weeks		112 (30.9%)
		TE once every 3 weeks		4 (1.1%)
	Anti-androgens	CPA 50	451 (100%)	
	Estrogens (76 missing)	Gel	4 (1.1%)	
		EV	219 (58.4%)	
		Patch 100 mcg/72 h	108 (39.2%)	
		Patch 50 mcg/72 h	12 (4.4%)	
		Patch 75 mcg/72 h	2 (0.7%)	
Number of TM reporting menstruation (%) (184 missing)				39 (16.4%)
Number of TM reporting spotting (%) (180 missing)				31 (12.8%)
Number of TM using contraceptives (%)				72 (15.9%)
Median Ferriman and Gallwey score				1.0 [0.0–3.0]
Serum estradiol levels (pg/mL)			25.3 [21.1–30.1]	36.9 [22.5–76.8]
Serum testosterone levels (nmol/L)			18.5 [14.0–23.3]	1.2 [0.9–1.6]
Serum LH levels (U/L)			3.9 [2.7–5.3]	4.5 [2.4–7.7]
Serum FSH levels (U/L)			3.5 [2.3–4.9]	5.6 [3.3–7.2]
Serum SHBG levels (ng/dL)			54.7 [35.0–81.0]	36.8 [26.7–51.3]
Median Utrecht Gender Dysphoria (UGDS) scores			22.0 [16.0–47.0]	29.0 [28.0–32.0]
Median quality of life (QOL) scores		Total	8.0 [7.0–9.0]	8.0 [7.0–9.0]
		Social	30.0 [26.0–33.0]	30.0 [26.0–34.0]
Median total SCL-90-R, Symptom Checklist scores (SCL-90R)			23.0 [8.0–47.3]	23.0 [10.0–54.0]

TW = transgender women, TM = transgender men, PA = PANAS positive affect scores, NA = PANAS negative affect scores, AG = androgen gel, TU = testosterone undecanoate, TE = testosterone esters, CPA = cyproterone acetate, EV = estradiol valerate, LH = luteinizing hormone, FSH = follicle stimulating hormone, SHBG = sex hormone binding globulin.

## Data Availability

The data presented in this study are available on request from the corresponding author. The data are not publicly available due to privacy restrictions.

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
