# Peer review of "Positive and Negative Affect Changes during Gender-Affirming Hormonal Treatment: Results from the European Network for the Investigation of Gender Incongruence (ENIGI)"

_jcm, 2021, doi:10.3390/jcm10020296_

Round 1
Reviewer 1 Report
The authors report on the first prospective study to look at the effect of hormone therapy on mood in transgender individuals. It is the work of the European Network for the Investigation of Gender Incongruence. This reports results from 4 sites in Europe. This cooperative study is needed, and the effort commended.
Specific comments
Are the hormone levels reported in the first table truly at baseline, as implied in the title of the table (testosterone is not reported)? If so, were the TM sampled at the same time of the menstrual cycle? Or are these levels on therapy? If so, they are not at goal as recommended in the Endocrine Society guidelines. Further, were 45% of TM not having menstrual cycles before therapy, or is this also post therapy?
Is the PANAS survey a validated survey or was it prepared for this study (this is not clear)
Is the Bradburn scale of psychological well being what you report as QoL in table 1 and the text (this is not clear)
General comments
There are a number of interesting findings reported here, and good, thoughtful discussion of some of these findings.
However, I have certain concerns that need to be addressed. The number of participants remaining in the study over time decreased significantly. At one year the population was down by 40% and by 18 months it was down by 85%. Hence, reporting results at 36 months when fewer than 5% of the population had data is problematic. Visual inspection of figure 1 would imply that at 1 year there was a difference from baseline.
I am also concerned that QoL survey was apparently only done at baseline. This is especially important given that you report some declines in positive affect scores in both TW and TM in the first year (see above point on the numbers), contrary to Fisher (ref 13).
You mention that there was no correlation with hormone levels and positive and negative affect scores. As mentioned above, it is not clear that Endocrine Society sex hormone goal levels were achieved in the study population.
Is there a possibility that there is some sort of bias created by which individuals filled out the questionnaire and those who didn’t (approximately half of study subjects)?
All TW received cyproterone. Could this have affected mood in some patients?
Author Response
Dear reviewer,
First of all we would like to thank you for your critical revision. We believe that the paper has been improved due to your comments. In this rebuttal we offer you a point-by-point response and explanation to your comments.
- Are the hormone levels reported in the first table truly at baseline, as implied in the title of the table (testosterone is not reported)? If so, were the TM sampled at the same time of the menstrual cycle? Or are these levels on therapy? If so, they are not at goal as recommended in the Endocrine Society guidelines. Further, were 45% of TM not having menstrual cycles before therapy, or is this also post therapy?
The hormone levels reported in the table are the baseline characteristics, before start of the gender-affirming therapy. These levels were independent of timing of menstrual cycle, measuring the levels at the same time during the cycle would be too difficult. In people without a menstrual cycle before testosterone therapy, this is due to the use of a progestogen agent.
We reported the testosterone levels in the table 1, they were indeed missing.
- Is the PANAS survey a validated survey or was it prepared for this study (this is not clear)
The PANAS survey is a validated survey. We added this in the paper (line 127-128).
- Is the Bradburn scale of psychological well being what you report as QoL in table 1 and the text (this is not clear)
The Bradburn scale (Life as a whole) is a questionnaire assessing general QoL and social QoL. We refer to this scale in table A1 and the text. We clarified this (line 163 and in the title of table A1).
- However, I have certain concerns that need to be addressed. The number of participants remaining in the study over time decreased significantly. At one year the population was down by 40% and by 18 months it was down by 85%. Hence, reporting results at 36 months when fewer than 5% of the population had data is problematic. Visual inspection of figure 1 would imply that at 1 year there was a difference from baseline. I am also concerned that QoL survey was apparently only done at baseline. This is especially important given that you report some declines in positive affect scores in both TW and TM in the first year (see above point on the numbers), contrary to Fisher (ref 13).
As the ENIGI study is an ongoing study, we decided to perform a data lock at certain time points, at which data were interpreted. This is why the number of participants decreases over time.
Affect is likely to contribute to someone’s QoL but is not the same. Affect are emotions we experience in response to certain situations. One can experience a positive and negative emotion at the same time, that’s why PA and NA are not opposites but intertwined. A high PA refers to a high state of energy, full concentration and pleasurable engagement. [1] As a lot of our emotions evolve from social interactions or situations, it is likely that PA is strongly influenced by our social situation. As we all know, transgender people experience a lot of social difficulties and these difficulties can worsen after initiation of gender-affirming hormonal therapy. In the study of Fisher [2] the social and legal indicators also worsened across time. This is a possible explanation why we found contrasting results in comparison to the study of Fisher. We added this consideration in the discussion (see line 287-289).
- You mention that there was no correlation with hormone levels and positive and negative affect scores. As mentioned above, it is not clear that Endocrine Society sex hormone goal levels were achieved in the study population.
Regarding the baseline LH and FSH levels we can conclude that the participants were not hypogonadal at baseline and the graphs that display the evolution of the testosterone in TM and estradiol concentrations in TW, show that the treatment goals as described by Hembree et al.[3], are rapidly achieved over time.
Graph 1: Prospective changes in median serum testosterone levels (nmol/L) in transgender men, shown as median with 95% confidence intervals.
Graph 2: Prospective changes in median serum estradiol levels (pg/mL) in transgender women, shown as median with 95% confidence intervals.
An important remark is that all TW received estradiol valerate of which the laboratory analyses is highly variable. Also the monitoring of serum estradiol levels in practice is to prevent supraphysiological levels. The treatment will not be adapted for a ‘lower’ estradiol level alone. This is in line with the recommendation, as there is no lower level recommendation for estradiol concentrations in TW, only a recommendation to not exceed the peak physiologic range.
In TM, on the other hand, there is a target range between 400-700 ng/dl (= 13.9 nmol/L - 24.3 nmol/L).
Endocrine Treatment of Gender-Dysphoric/ Gender-Incongruent Persons: An Endocrine Society Clinical Practice Guideline. JCEM 2017.[3]
- Is there a possibility that there is some sort of bias created by which individuals filled out the questionnaire and those who didn’t (approximately half of study subjects)?
We do not know the exact reason why a large part the participants did not fill out the questionnaire anymore after a certain amount of time. As mentioned above, we think that participants find the questionnaires time-consuming and we can imagine that there is also a repetition bias. Participants have to answer the questions every three months and probably get ‘bored’ by it. We mentioned this limitation in the discussion (see line 362-364).
- All TW received cyproterone. Could this have affected mood in some patients?
Treatment with Cyproterone may occasionally lead to mood disorder problems. At this point it is unclear which is the underlying mechanism.[4] Previous trials have shown no significant difference in psychological well-being when comparing Cyproterone acetate with a synthetic gonadotropin-releasing hormone.[5] There are 451 TW in the study who all receive cyproterone acetate. It seems unlikely that the low frequency of such event influenced the results in this large cohort.
References
- Watson, D.; Clark, L.A.; Carey, G. Positive and Negative Affectivity and Their Relation to Anxiety and Depressive Disorders. J. Abnorm. Psychol. 1988, doi:10.1037/0021-843X.97.3.346.
- Fisher, A.D.; Castellini, G.; Ristori, J.; Casale, H.; Cassioli, E.; Sensi, C.; Fanni, E.; Amato, A.M.L.; Bettini, E.; Mosconi, M.; et al. Cross-sex hormone treatment and psychobiological changes in transsexual persons: Two-year follow-up data. J. Clin. Endocrinol. Metab. 2016, doi:10.1210/jc.2016-1276.
- Hembree, W.C.; Cohen-Kettenis, P.T.; Gooren, L.; Hannema, S.E.; Meyer, W.J.; Murad, M.H.; Rosenthal, S.M.; Safer, J.D.; Tangpricha, V.; T’Sjoen, G.G. Endocrine treatment of gender-dysphoric/ gender-incongruent persons: An endocrine society∗clinical practice guideline. J. Clin. Endocrinol. Metab. 2017, doi:10.1210/jc.2017-01658.
- Cheung, A.S.; Wynne, K.; Erasmus, J.; Murray, S.; Zajac, J.D. Position statement on the hormonal management of adult transgender and gender diverse individuals. Med. J. Aust. 2019, doi:10.5694/mja2.50259.
- Gava, G.; Cerpolini, S.; Martelli, V.; Battista, G.; Seracchioli, R.; Meriggiola, M.C. Cyproterone acetate vs leuprolide acetate in combination with transdermal oestradiol in transwomen: a comparison of safety and effectiveness. Clin. Endocrinol. (Oxf). 2016, doi:10.1111/cen.13050.
Reviewer 2 Report
This manuscript presents results of prospective data over 3 years on affect measured by the PANAS questionnaire following gender affirming hormones in the ENIGI study. I applaud the authors in attempting to measure the psychological impact of gender affirming hormones, however I think the presentation of this manuscript is confusing and it was difficult to follow. The prospective nature over 3 years is a strength and I think the presentation of the cross-sectional data with comparisons to some of the prospective measures is confusing and should be reconsidered. The major limitation is the lack of a control group or presentation of normative data which makes the findings difficult to interpret.
Introduction:
- I would suggest including hypotheses and rationale for your hypotheses and more clearly emphasising the aims.
Methodology:
- Affect is measured in this study by the self-reported PANAS questionnaire, but other measures of quality of life, gender dysphoria and psychological burden were not measured prospectively over time. Why was the psychological battery only measured at baseline?
- What is the general population median PANAS score? How does it compare to other groups?
- Why compare cross-sectionally between TW and TM at baseline? There is no clear hypothesis for this or rationale. You assert that this is novel as it is a prospective study and I think that it is clearer and less confusing to present just the prospective data. I would suggest deleting 3.2 altogether and deleting the cross-sectional analyses in 3.3 and 3.4.1. It doesn't seem to add anything to the paper or to the clinical interpretation and is confusing. W
Results:
- The term "control visit" is misleading. I am anticipating a control group however I would suggest rewording, perhaps to "follow-up visit" or something similar.
- Regarding figures, it is difficult to see the data for TW and TM as the graphs overlap. Can this be made clearer? The figure legend needs to explain what is depicted i.e. the points are the median and bars are interquartile range?
- The PANAS data in the figures looks highly variable over time and authors should be wary of interpreting specific time points without a control group.
- Can you present the testosterone concentrations over time for TM and the estradiol concentrations over time for TW? Does this explain the lack of PA?
- Whilst you measured Ferriman Gallwey score as an attempt to objectively measure masculinisation, did you have any objective measures of feminisation i.e. breast circumference? Does this explain the lack of PA?
- Section 3.1, I would suggest some subheadings i.e.
Positive Affect, then Negative Affect OR Trans Women then Trans Men. It is hard to follow.
- It is stated in lines 230 - 231 that "The three PA as well as NA groups (increasing, stable or decreasing) did not differ regarding Ferriman and Gallwey scores" but then in lines 245-246 that "Prospective changes in PA were correlated to prospective changes in total Ferriman and Gallwey scores in TM (ρ=0.910, P=0.002)." Can you explain this?
Discussion:
- The discussion overinterprets the findings given that there is no control group and small variations despite being statistically significant are not consistent over time. Inference is limited. Significant changes could for example be explained by regression to the mean in addition to the possible explanations outlined by the authors. I think the main message is that affect is complex in transgender people and that PANAS does not significantly change over time (or that PANAS may not necessarily be useful tool to measure).
- The limitations of PANAS need to be explained further.
- The results in fact may be used by anti-trans groups to advocate against providing gender-affirming hormone therapy and the limitations of this study methodology need to be strongly highlighted throughout.
Conclusions:
Conclusions need to be revised so as not to overinterpret the uncontrolled data. A possible suggestion is: This uncontrolled prospective study found that PA and NA did not appear to change significantly from baseline to 3 years follow up after starting gender affirming hormone therapy. Variability during follow up suggests that affect is complex and we were unable to measure this complexity or reasons for increases or decreases in affect in this study. Changes in PA and NA were not associated with ferriman gallwey scores, menstruation or surgery....
Author Response
Dear reviewer,
First of all we would like to thank you for your critical revision. We believe that the paper has been improved due to your comments. In this rebuttal we offer you a point-by-point response and explanation to your comments.
- I would suggest including hypotheses and rationale for your hypotheses and more clearly emphasising the aims.
As suggested, we emphasized our hypothesis and subsequently the aims of our study (lines 71-79).
- Affect is measured in this study by the self-reported PANAS questionnaire, but other measures of quality of life, gender dysphoria and psychological burden were not measured prospectively over time. Why was the psychological battery only measured at baseline?
Thank you, we agree on this comment. Looking back on the study protocol it would have been better to also measure the psychological battery in a prospective way. But at baseline we thought that this would be too time-consuming for the study participants. We highlighted this limitation in our discussion (line 364-369).
- What is the general population median PANAS score? How does it compare to other groups?
Crawford and Henry, British Journal of Psychology, 2010 [1]
As you can see the median scores of the transgender population are similar to the median scores of the general population in the UK (1001 participants).
- Why compare cross-sectionally between TW and TM at baseline? There is no clear hypothesis for this or rationale. You assert that this is novel as it is a prospective study and I think that it is clearer and less confusing to present just the prospective data. I would suggest deleting 3.2 altogether and deleting the cross-sectional analyses in 3.3 and 3.4.1. It doesn't seem to add anything to the paper or to the clinical interpretation and is confusing.
We agree with your point of view. The assessment of the cross-sectional data is confusing. Therefore, we removed this data out of section 3.2 and 3.3. We decided to maintain the cross-sectional analyses of 3.4.1 (the psychological correlates) because these data can be a guidance for further research, since the results of the prospective analysis are rather poor.
- The term "control visit" is misleading. I am anticipating a control group however I would suggest rewording, perhaps to "follow-up visit" or something similar.
As suggested, we changed control visit to follow-up visit in the entire paper.
- Regarding figures, it is difficult to see the data for TW and TM as the graphs overlap. Can this be made clearer? The figure legend needs to explain what is depicted i.e. the points are the median and bars are interquartile range?
We agree and have updated the graphs (figure 1 and 2). They show median + error bars (95%CI). You can also notice that we chose to change the scales on the NA questionnaire to 10-20, because this is a more truthful representation of the changes whilst still remaining visible.
- The PANAS data in the figures looks highly variable over time and authors should be wary of interpreting specific time points without a control group.
The data are indeed highly variable over time because the PANAS is self-reported and it also reports the affect of the person on a specific moment and not over a longer period of time. As mentioned in our discussion this may include some bias. This bias could be reduced by using a control group. We did not incorporate a control group in our study because it would not be ethically desirable to deny this control group gender-affirming hormonal therapy. We tried to reduce the bias by using a large cohort in a real-life setting with no exclusion of certain groups. Of course, we agree that caution is needed in interpreting the data. We therefore adjusted our aims in the introduction.
- Can you present the testosterone concentrations over time for TM and the estradiol concentrations over time for TW? Does this explain the lack of PA?
We did not find any correlation between PANAS PA scores and sex steroid concentrations nor we did we find at baseline and during follow-up we a significant difference between TM an TW. These findings suggest that influence of testosterone or estrogens is similar on affect (lines 296-298). Regarding the baseline LH and FSH levels we can conclude that the participants were not hypogonadal at baseline and as you can see in the next figure, the treatment goals as described by Hembree et al. [2] are rapidly achieved over time.
The testosterone and estradiol concentrations over time are shown in the 2 graphs below.
Graph 1: Prospective changes in median serum testosterone levels (nmol/L) in transgender men, shown as median with 95% confidence intervals.
Graph 2: Prospective changes in median serum estradiol levels (pg/mL) in transgender women, shown as median with 95% confidence intervals.
- Whilst you measured Ferriman Gallwey score as an attempt to objectively measure masculinisation, did you have any objective measures of feminisation i.e. breast circumference? Does this explain the lack of PA?
Thank you for this remark. From previous research we know that breast development may not always live up to the expectations.[3] Breast circumference in itself is a suboptimal measure of feminisation, and a more detailed evaluation is needed to include it as an objective measure.[4] Therefore we chose not to use breast circumference in this analysis. Unfortunately, the 3D evaluation is not available in this cohort. We added this limitation in the discussion (line 357-358).
For now, we only found a prospective positive correlation between the Ferriman Gallwey Score and PA in TM. When comparing the subgroups (PA increasing, decreasing or stable) we did not find a difference in the Ferriman Gallwey Score due to small sample size in one of the groups (line 321-323).
- Section 3.1, I would suggest some subheadings i.e. Positive Affect, then Negative Affect OR Trans Women then Trans Men. It is hard to follow.
Thank you for this suggestion, we applied subheadings in this section (PA, NA and subgroup analysis).
- It is stated in lines 230 - 231 that "The three PA as well as NA groups (increasing, stable or decreasing) did not differ regarding Ferriman and Gallwey scores" but then in lines 245-246 that "Prospective changes in PA were correlated to prospective changes in total Ferriman and Gallwey scores in TM (ρ=0.910, P=0.002)." Can you explain this?
We agree that this seems strange, but this is due to sample size. We could leave the comparison between the three groups out, to make the manuscript less confusing. For now, we added this explanation to the discussion (see line 321-323).
- The discussion overinterprets the findings given that there is no control group and small variations despite being statistically significant are not consistent over time. Inference is limited. Significant changes could for example be explained by regression to the mean in addition to the possible explanations outlined by the authors. I think the main message is that affect is complex in transgender people and that PANAS does not significantly change over time (or that PANAS may not necessarily be useful tool to measure).
Thank you for this remark, we agree that the results should be considered with caution since the absolute values are small. We adjusted our discussion and conclusion to a more nuanced message (line 385 - 394). The lack of a control group is mentioned as a limitation in our discussion (line 356 - 357).
- The limitations of PANAS need to be explained further.
We agree that we should definitely include the limitations. They are stated in lines 358 – 366.
- The results in fact may be used by anti-trans groups to advocate against providing gender-affirming hormone therapy and the limitations of this study methodology need to be strongly highlighted throughout.
We highlighted the repetition bias as well as the need for other questionnaires to also evaluate minority stress, gender dysphoria and interpersonal relationships. (line 366-371)
- Conclusions need to be revised so as not to overinterpret the uncontrolled data. A possible suggestion is: This uncontrolled prospective study found that PA and NA did not appear to change significantly from baseline to 3 years follow up after starting gender affirming hormone therapy. Variability during follow up suggests that affect is complex and we were unable to measure this complexity or reasons for increases or decreases in affect in this study. Changes in PA and NA were not associated with ferriman gallwey scores, menstruation or surgery....
We incorporated your suggestion and nuances in our discussion and conclusion. In the entire discussion and conclusion we replaced the terms PA and NA by PANAS NA and PA scores to underscore that nuance.
References
- JR, C.; JD, H. The Positive and Negative Affect Schedule (PANAS): construct validity, measurement properties and normative data in a large non-clinical sample. Br. J. Clin. Psychol. 2004.
- Hembree, W.C.; Cohen-Kettenis, P.T.; Gooren, L.; Hannema, S.E.; Meyer, W.J.; Murad, M.H.; Rosenthal, S.M.; Safer, J.D.; Tangpricha, V.; T’Sjoen, G.G. Endocrine treatment of gender-dysphoric/ gender-incongruent persons: An endocrine society∗clinical practice guideline. J. Clin. Endocrinol. Metab. 2017, doi:10.1210/jc.2017-01658.
- De Blok, C.J.M.; Klaver, M.; Wiepjes, C.M.; Nota, N.M.; Heijboer, A.C.; Fisher, A.D.; Schreiner, T.; T’Sjoen, G.; Den Heijer, M. Breast development in transwomen after 1 year of cross-sex hormone therapy: Results of a prospective multicenter study. J. Clin. Endocrinol. Metab. 2018, doi:10.1210/jc.2017-01927.
- de Blok, C.J.M.; Dijkman, B.A.M.; Wiepjes, C.M.; Staphorsius, A.S.; Timmermans, F.W.; Smit, J.M.; Dreijerink, K.M.A.; den Heijer, M. Sustained breast development and breast anthropometric changes in three years gender-affirming hormone treatment. J. Clin. Endocrinol. Metab. 2020, doi:10.1210/clinem/dgaa841.
Round 2
Reviewer 2 Report
Thank you for the revisions. The authors have adequately addressed my comments.